# What impact has the NIHR Academic Clinical Fellowship (ACF) scheme had on clinical academic careers in England over the last 10 years? A retrospective study

Sally Clough,[1] James Fenton,[1] Helen Harris-Joseph,[1] Leesa Rayton,[1] Caroline Magee,[1] David Jones,[2] Lisa Ann Cotterill,[1] James Neilson[3]

► Prepublication history and additional material are available. To view these files please visit the journal online (http://dx.doi.org/ 10.1136/ bmjopen-2016-015722).

[1]NIHR Trainees Coordinating Centre, Leeds Innovation Centre, Leeds, UK
[2]NIHR Newcastle Biomedical Research Centre and Institute of Cellular Medicine, Newcastle University, Newcastle upon Tyne, UK
[3]Department of Women's and Children's Health, The University of Liverpool, Liverpool, UK

**Correspondence to**
Dr Sally Clough;
sally.clough@nihr.ac.uk

## ABSTRACT

**Objectives** The Academic Clinical Fellowship (ACF) was introduced to support the early career clinical and research training of potential future clinical academics in England. The driver for the model was concern about falling numbers of clinical academic trainees. This study examines the impact of the ACF model, over its first 10 years, in developing clinical academic careers by tracking the progression of ACF trainees.

**Design** Retrospective analysis of National Institute for Health Research (NIHR) ACF career progression. This was performed using mixed methods including routine data collections of career destination, analysis of application rates to doctoral level fellowships and supplemented by survey information that captured the perceived benefits and challenges from previous ACFs and their current career activities.

**Participants** 1239 NIHR ACFs who completed or left their posts between 2006 and March 2015.

**Results** ACFs are perceived by the candidate population as attractive posts, with high numbers of applications leading to high fill rates. Balancing clinical and academic commitments is one of the reported challenges when completing an ACF. We have found that undertaking an ACF was shown to increase the likelihood of securing an externally funded doctoral training award and the vast majority of ACFs move into academic roles, with many completing PhDs. Previous ACFs continue to show positive career progression, predominantly in translational and clinical research. The knowledge acquired during the ACF continues to be useful in subsequent roles and trainees would recommend the scheme to others.

**Conclusions** The NIHR ACF scheme is successful as part of an integrated training pathway in developing careers in academic medicine and dentistry.

## INTRODUCTION

The Integrated Academic Training (IAT) Programme was launched in October 2005 and became a flagship scheme of the National Institute for Health Research (NIHR) when it was established in 2006. This built on

### Strengths and limitations of this study

► This is the first cohort analysis of National Institute for Health Research (NIHR) Academic Clinical Fellowship (ACF) career progression, providing evaluation of a novel career development model.

► With over 1400 completed ACF posts and 10 years since the NIHR Integrated Academic Training pathway was introduced, it is timely to conduct this analysis.

► The first career destination of previous ACFs were collected from their host partnerships, with known destinations reported for 83% of the cohort (17% unknown or unreported).

► We have not measured the career progression of clinicians who have not completed an ACF but have entered clinical academia via an alternative route.

► Analysis of a follow-up survey is presented as illustrative information as response rates (40%) were limited by a lack of up-to-date contact information.

recommendations published in a report[1] of the Academic Careers Sub-Committee of Modernising Medical Careers and The UK Clinical Research Collaboration in March 2005, which identified the lack of a clear career structure as a barrier to junior doctors and dentists being able to establish themselves in academia alongside developing their clinical careers. The resultant IAT pathway combines academic and clinical training and incorporates the clinician scientist positions recommended and implemented from the earlier Savill report.[2]

The NIHR Academic Clinical Fellowship (ACF) was created as a new career development scheme and is the first step on the NIHR IAT pathway and provides predoctoral academic training during the specialty training period for doctors and dentists. The

ACF may be completed over a maximum of 3 years' full time or 4 years for general practitioners and general dental practitioners, providing 25% of protected time over the course of the post for developing research skills alongside clinical training.

The NIHR IAT Programme supports approximately 250 medical and 20 dental NIHR ACFs each year in England. Post are allocated on an annual basis to local IAT partnerships of Health Education England (HEE), higher education institutes, medical schools and NHS Trusts/organisations. Funding supports local hosting and management of the posts, covering basic salary, access to a local, formal academic research training programme, which provides general research skills, and a yearly £1000 bursary for the trainee to attend scientific meetings and conferences.

The successful endpoint of an NIHR ACF is considered to be a funded application to undertake a research training award either immediately or later, depending on individual circumstances such as continuing in clinical training before pursuing a research training award. Trainees who decide to leave the academic pathway join an alumni programme that aims to support research studies elsewhere in the NIHR.

To date, 2247 ACF posts have been funded by the NIHR since the scheme was launched, and over 40 additional, locally funded, ACF posts have also contributed to the programme. The scheme has evolved over time, with flexible entry level points that range from specialty or core training year 1 (ST1 or CT1) following foundation level training to ST3, for all General Medical Council or General Dental Council specialties. Entry at ST4 is also permitted for the six psychiatric specialities, paediatrics and emergency medicine posts. The entry level and the specialty is determined locally by the IAT partnerships who undertake recruitment following NIHR guidance and national recruitment processes.

It is 10 years since NIHR ACFs were first advertised, and this study is the first published in-depth evaluation of career progression of the NIHR ACF post holders. The aim of the study was to determine whether the ACF programme has improved the access to clinical academic careers by providing clinicians with early career research training, enabling progression along a clinical academic pathway.

## METHODS

Information about NIHR ACF career progression has been collated from a number of sources. The first known academic career destinations of ACFs have been collected from host partnerships since 2006. NIHR application data were analysed to inform the progression of ACFs to PhD training positions, and success rates were compared using a two-tailed Z test to compare proportions. Gender information held by NIHR and collected from HEE on specialty post holders was analysed using a $\chi^2$ goodness-of-fit test and a two-tailed Z test to compare

proportions. For statistical tests, the significance threshold was set at .05.

Additionally, a retrospective online survey was sent directly to individuals who had completed their awards in order to capture further details. The online survey was targeted at trainees who had been awarded NIHR ACFs since April 2006 and who had subsequently completed or left their award up to mid-March 2015, indicated by data collected from HEE Local Education and Training Boards (formerly postgraduate deaneries). Internet searches supplemented contact details already held by NIHR to provide email addresses for previous ACFs. However, due to the mobile nature of trainees, we know contact emails are changed on a regular basis, and we cannot confirm the survey reached all the intended recipients. Open-text answers were evaluated using thematic analysis.

## RESULTS

Since the scheme was launched, 2247 ACF posts have been funded by the NIHR. Of these trainees, 1239 had completed their ACF or left their post by the audit date and were included in the analysis.

### Application and fill rates

Information on application rates for non-ACF doctors and dentists is currently being collected, but early indications suggest that NIHR ACF posts are highly competitive and attract a large number of applicants and will be the focus of a future study. For general practice, on average, over five applications per post have been received for ACFs in each round between 2011 and 2015. This compares to less than two applications that were made per GP ST1 post available in the same time period (data provided by GP National Recruitment (HEE).

Subsequent fill rates of ACF posts are high and compare favourably with annual standard specialty recruitment figures (generally above 89% and now at a steady fill rate of 96% since 2013, supplementary figure 1). The proportion of females in a current ACF post was 48%, and the proportion of males was 52%. This difference was not statistically significant (approximately 750 posts). The equal gender distribution was also apparent for those who had completed ACF posts at the time of the audit (46% female and 54% male (over 1400 posts)). This balanced profile is however significantly different (p=0.0007) to the proportion of females in specialty trainee posts (data courtesy of HEE, November 2015) where a significantly greater number of trainees were female (57% of 11 160 current medical trainees *excluding ACFs, out of programme, locum appointments for training and foundation trainees, p<0.0001).

### First destinations of ACFs

Annual and quarterly data returns from the ACF host organisational partnerships have provided known destination information for 83% of all previous ACF post holders. These first career destinations of NIHR ACFs indicate that, overall, 47% of ACFs progress to a

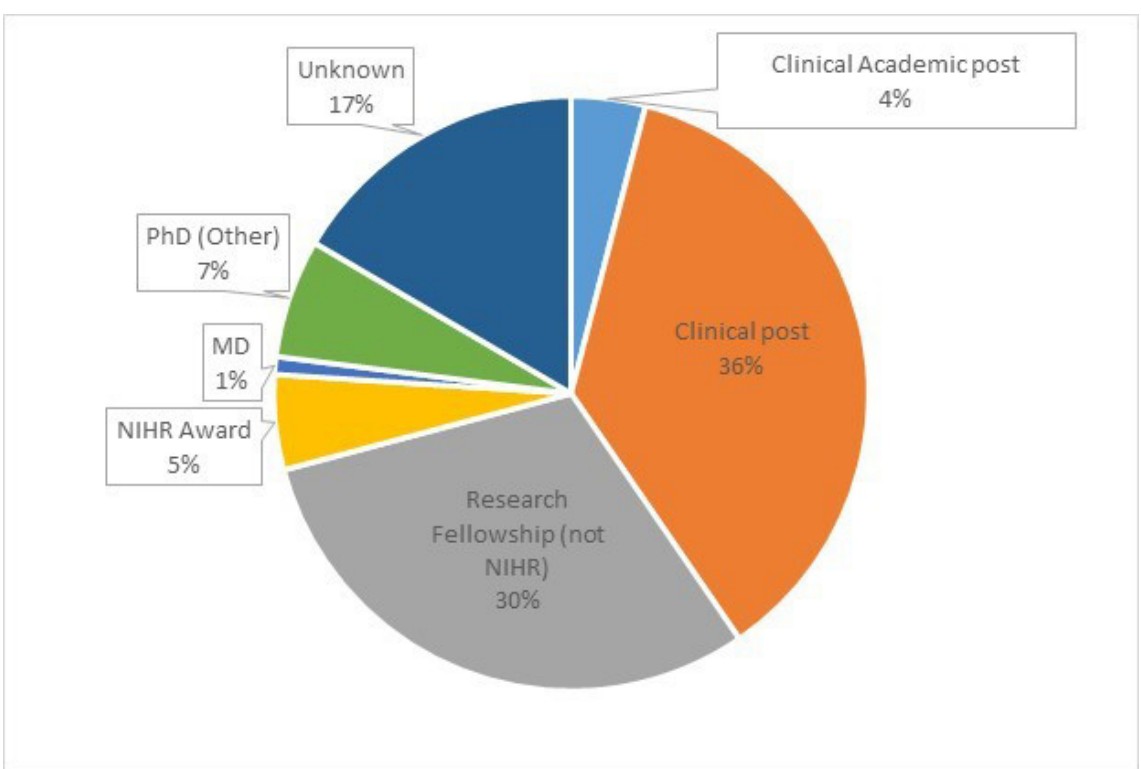

**Figure 1** First career destinations following an NIHR Academic Clinical Fellowship (ACF) (n=1239). Information collected from ACF host institutions. NIHR, National Institute for Health Research.

position with an academic component and 36% go into a clinical post immediately after their ACF (figure 1).

### PhD funding application success rates

Overall, individuals who have held an ACF are significantly more successful than their specialty trainee colleagues when applying for an NIHR Doctoral Research Fellowship (DRF), which is open to both clinically and non-clinically qualified individuals to undertake a PhD. This has been the case since 2010. Of 190 ACF applicants, 54 were successful (28%) compared with 51 out of 273 (19%) of non-ACF clinical applicants (p=0.01). The overall success rate from 2010 to 2015 was 19% (194/1001). Success rates are also higher compared with all applicants to the NIHR DRF scheme, regardless of professional background (all non-ACFs: 17%, n=811, p=0.00005). Typically, about 35 NIHR DRFs are awarded each year. This increased success has also been observed by the Wellcome Trust for their Research Training Fellowships (RTFs) and a high proportion of successful Medical Research Council RTFs are previous NIHR ACF holders (personally communicated data from each funder). Overall, the number of ACFs subsequently undertaking a PhD or higher level research training (for those who may already hold a PhD) is encouraging. Data collected from institutional data includes 43% that immediately go on to study a further research training award funded by NIHR or other funders.

### Online survey

Contact details were available for 1119 (90%) of the trainees who had left or completed their ACF and were invited to participate in a follow-up online survey.

Fifty emails were undeliverable and 433 survey responses were received; a 40% response rate. Forty-one per cent of responders were female, 58% male and 1% responders preferred not to specify their gender (n=419). Although this gender profile differs slightly from the more equal split of all previous ACFs, there was a similar specialty spread and comparable profile of the first destinations of survey respondents to that in the data collected from institutions; however, due to the lower response rate, we have used the responses only as illustrations of career paths and of reported enablers and barriers to the clinical academic pathway.

### Benefits and challenges

The survey asked participants what the main benefit was from undertaking an NIHR ACF, and a number of factors were described by survey respondents (box 1). These included protected research time, the use of an ACF as an access point to clinical academic careers and experience in research, the practical research skills gained and the networking and support opportunities that were available.

Balancing clinical and academic commitments was described by almost half of responders to be the main challenge they faced (box 1). Other challenges

<table>
<tr><td>

**Box 1   The main benefits and challenges of undertaking an ACF reported by previous trainees**

**Benefits**
**Protected time**
'*Gave me direct quality time in the laboratory to pursue my research, and act as a springboard to my Research Training Fellowship application*'
'*Exploring options and being able to identify the research question/area that I am passionate about*'
**Gateway to research**
'*I learned a huge amount! Really allowed me to think about the wider issues that impact on clinical practice. I really loved it and I has taught me a lot.*'
'*springboard to career as clinical academic (now post-doc CL)*'
'*The opportunity to experience research first hand and develop an interest to lead onto a higher degree*'
**Research skills**
'*Knowledge of research methods and academic writing*'
'*Allowed me to undertake research projects and greatly improved my writing skills for publication and abstract submissions*'
**Networking and support**
'*Time to spend developing relationships in research and teaching.*'
'*Being part of established research network*'
**Challenges**
**Balancing clinical and academic activities**
'*Organising protected academic time and balance that with service requirement*'
'*Satisfying both clinical and academic requirements for training/career progression in parallel*'
**Organisational support**
'*Clinicians see you as a part time trainee as you do academic training. It is not always easy to get them to treat you in the same way. Clinicians presume that you are not as skilled clinically as you spend more time doing research.*'
'*Persuading the trust to allow me to take research time*'
**Personal**
'*Deciding upon future career and funding applications*'
'*Identifying research question/program of work to progress to next tier*'
**Financial**
'*Obtaining competitive funding for subsequent post-doctoral research*'
'*Obtaining PhD funding in time to start immediately following end of ACF*'

</td></tr>
</table>

highlighted included financial challenges, which referred to finding funding for the next step beyond the ACF as well as funding for research within the ACF and consideration given to salary differences when undertaking an ACF. Organisational support and a number of personal decisions were also raised as challenges. Some of these challenges were also reflected when we asked those who were not currently in a role with an academic component what the main decision was not to continue at this stage with an academic career. A variety of factors were cited, including: needing to continue or maintain clinical competencies, work/life balance, financial, lack of available opportunities and poor institutional support. Of these respondents, there was a further indication that they thought it was likely that they would return to an academic position in

the future and that the current role was a part of their career progression.

A small proportion of both male and female participants reported that they did not have any challenges when completing their ACF.

## DISCUSSION

The introduction of the NIHR ACF in England has provided a visible and attractive route into clinical academic training and inspired a new generation of doctors and dentists to follow an academic career. Fill rates have been consistently high since the beginning. The post itself provides an opportunity to combine clinical and academic training and supports the trainee to prepare a competitive application for a nationally funded RTF. With over 2200 ACFs funded in England since 2006, the scheme has clearly increased the pool of trainees working at this level. The posts are attractive with application numbers that generally exceed those for standard specialty training posts.

The presented work is the first analysis of current career outcomes of NIHR ACF post holders, which shows that the majority of post holders are using an ACF to access the available clinical academic pathway. We have found that the ACF programme, as an intervention, is successful in its aim to enable clinicians to competitively apply for and obtain PhD positions after the ACF post. Typically, ACFs are twice as likely to have a successful NIHR DRF application than their clinical colleagues who have not held an ACF award. This higher success rate is also reflected in applications for other externally funded fellowships. From institutional data, up to 43% of ACFs go on to a PhD or further research fellowship immediately after their post.

It also follows that a number of the 36% of NIHR trainees who moved straight to a clinical post after their ACF can and will move into an academic post at a later date. The clinical post may have suited their situation whether personal or professional at the time, or they simply may not have been successful with an initial fellowship application but are successful at a later date. Further tracking has shown that there are clear opportunities to continue to engage with the clinical academic pathway, demonstrated by a number of survey respondents who initially held a clinical role who then transition into further academic training opportunities, particularly PhD and clinical lecturer (post-doctoral) training. In addition to formal research training awards, a number of other outcomes would be considered successful. ACFs that do not continue onto a formal clinical academic training path and decide to return to clinical posts do so with a greater understanding of research and new skills from their ACF experience. Examples from the follow-up survey include those who indicate they are in a clinical role but also that they undertake some research activities as part of that role. Others indicated no formal time within their current role for research activities but

specified that they carry out peer review and supervision of PhD students.

It is clear that the NIHR ACF programme is providing clinicians with a route to clinical academic roles and/or with the skills they need to contribute to research activity within their current roles, providing a pool of research aware and trained clinicians. This reflects the reported intentions of the 2008 cohort of ACFs who declared they intended to work in clinical academic posts or clinical posts with some aspect of research in an earlier survey of ACF trainees by Goldacre *et al*.[3]

It should also be stated that the NIHR pathway is not the only route to an academic career; other paths are possible both in England and the rest of the UK,[4] where similar posts are available.

The follow-up survey has its limitations due to the low response rate, but it did provide insights into some of the benefits and challenges faced by those entering the clinical academic pathway. Benefits to undertaking ACF that were reported by previous trainees included the research experience they gained, protected time to prepare for the next step (eg, a competitive fellowship) and networking to facilitate finding supervisors. The follow-up survey also highlighted a number of challenges reported by ACFs regarding undertaking their award. These included financial concerns, a lack of support from both institutional and personal (supervisor and peers) sources and the challenge of balancing clinical and academic commitments. Such issues are commonly cited in relation to academic training,[3 5] and our survey responses indicate they are experienced at this early career stage. Despite perceived challenges around academic dentistry, the results of a recent survey[4] of dental academic trainees were very positive and reported that a majority of dental academic trainees would recommend an academic career to their peers.

Looking to the future, NIHR will continue to monitor and track ACF post holders along their chosen pathways and careers and evaluate the exciting impact of these posts on both clinical academic medicine and dentistry. This review and the ongoing evaluation will provide information to ensure the provision of an evidence-based trainee policy for NIHR. ACFs clearly face a range of challenges, particularly around the need to balance research, training and clinical service pressures. NIHR is therefore working with its partners to set out key principles and obligations for those in receipt of funding to ensure that high-quality clinical academic training is provided. NIHR have prepared a new NIHR Guide to Integrated Academic Training with clear guidance of expectations and processes for both host partnerships and trainees in order to support ACFs in the programme. As has been outlined by other commentators,[6] the impact over longer time periods will be required to continue to evaluate this important programme; however, this 10-year analysis clearly demonstrates the success of the programme of starting a significant number of clinicians on the clinical academic track.

**Acknowledgements** We would like to thank all of the survey respondents, Professor Linda McGowan for advice on the project and reviewing the manuscript, Professor Jenny Hewison for reviewing the manuscript and Dawn-Marie Burgess for data support. We would also like to thank the following for sharing data with us: HEE, particularly Jonathan Howes, Wellcome Trust and the Medical Research Council.

**Contributors** LAC, CM and JN conceived the idea and designed the study. HH-J and LR obtained and analysed institutional data. SC designed the survey and collected and analysed the data. SC and JF drafted the manuscript. JF oversaw institutional data collection. DJ critically reviewed the manuscript. LAC and CM revised the draft manuscript. All authors read and approved the final manuscript.

**Funding** No specific funding to report. Authors are representatives of the funding body NIHR.

**Competing interests** SC, JF, HH-J, LR, CM and LAC work for the NIHR Trainees Coordinating Centre as employees of the Leeds Teaching Hospitals NHS Trust, which receives funding from the Department of Health. DJ declares that he acts as NIHR Dean for Faculty Trainees and receives financial payment from the Department of Health for this role. JN declares no competing interests.

**Provenance and peer review** Not commissioned; externally peer reviewed.

**Data sharing statement** No additional data available.

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
