## [Reviewer comments · BMJ Open]

ARTICLE DETAILS

TITLE (PROVISIONAL)	What impact has the NIHR Academic Clinical Fellowship (ACF) scheme had on clinical academic careers in England over the last 10 years?: A retrospective study.
AUTHORS	Clough, Sally; Fenton, James; Harris-Joseph, Helen; Rayton, Leesa; Magee, Caroline; Jones, David; Cotterill, Lisa; Neilson, James

VERSION 1 - REVIEW

REVIEWER	Paul Stewart University of Leeds Chair MRC Training & Careers Group
REVIEW RETURNED	30-Jan-2017

GENERAL COMMENTS	This is the first report of outcome data from the NIHR Academic Clinical Fellowship scheme (ACF) first introduced in 2005-2006. It is timely to share outcome data from the first 10 year experience. The data suggest the scheme is indeed capacity building clinical academic careers across England (as assessed by the single metric of progression to NIHR funded doctoral research fellowships) but it is a real shame (and hopefully a lesson for the future) that the scheme was not configured with appropriate outcome milestones to be in place at inception. If so then we would have a suitable control group (unsuccessful ACF applicants) and perhaps more robust data on other key outcomes including progression to RCUK and AMRC CRTFs, HEFCE and NIHR Clinical Lectureships rather than just the 43% progression to immediate PhD and 36% straight to clinical post. The authors recognise this major limitation of the study in the abstract and discussion. Minor points: 1. Page 4 line 38. "Is not significantly different" from what? The point being there was no gender preference. Please re-write2. Page 4 line 41 - delete "extremely" significant. As this is a statistical analysis please give numbers of the HEE specialty trainee posts sampled.3. Page 4 line 58 - again the total denominator data would be useful. Please give numbers of ACF's and specialty trainees applying and numbers appointed. As this is the first report a breakdown of specialty for the ACF's would be helpful.4. Page 5 lin 17. The survey isn't robust and doesn't really provide additional data to that reported in reference 7 which as I recall was a cross-funder containing NIHR fellows albeit at a later stage in the career (CRTF and Clinician Scientist). This might be discussed in more detail.5. Page 7 line 30. I didn't really understanding this paragraph at all. If the destination is purely clinical how does this in anyway reflect the
--

	current debate as to gender balance for Academic careers? Without data on onward career progression of this "immediate clinical" group I think this paragraph is redundant.
--	---

REVIEWER	Jane M Blazeby University of Bristol, UK
REVIEW RETURNED	18-Feb-2017

GENERAL COMMENTS	1. It is not clear why this paper only focuses on the impact of the ACF post and it neglects to examine ACL and AF2 posts - all of which were introduced to improve the number and quality of clinical academics. The paper also incorrectly says that an ACF post is the first step on the NIHR pathway - however the AF1/2 posts are the first step 2. The abstract could benefit from some details of what the survey was aiming to assess - and its purpose - 3. The results in the abstract seem to go beyond what was likely to be surveyed - e.g. numbers of applicants and successful appointments - so the methods are unclear in this summary abstract 4. It is unclear what a 'successful ACF is defined to be award of PhD funding'. Another key success could be that the candidate is made aware of the clinical academic world and they decide its not for them and they return to full time NHS posts - I think that more thought into some of this is needed 5. Is it possible that there is responder bias (i.e. ACFs that are successful with PhD applications have responded more than those not successful) 6. One of the main challenges identified by ACFs of this post is the balance between clinical and academic work - I think that the discussion section of the paper could be improved if the authors considered how to tackle these issues, in particular, how they may work with NHS consultants and training pathways to make this more positive and how ACFs can be supported to negotiate these difficult issues
--

VERSION 1 – AUTHOR RESPONSE

Reviewer: 1

Paul Stewart

University of Leeds

Please state any competing interests or state 'None declared': Chair MRC Training & Careers Group

This is the first report of outcome data from the NIHR Academic Clinical Fellowship scheme (ACF) first introduced in 2005-2006. It is timely to share outcome data from the first 10 year experience. The data suggest the scheme is indeed capacity building clinical academic careers across England (as assessed by the single metric of progression to NIHR funded doctoral research fellowships) but it is a real shame (and hopefully a lesson for the future) that the scheme was not configured with appropriate outcome milestones to be in place at inception. If so then we would have a suitable control group (unsuccessful ACF applicants) and perhaps more robust data on other key outcomes including progression to RCUK and AMRC CRTFs, HEFCE and NIHR Clinical Lectureships rather

than just the 43% progression to immediate PhD and 36% straight to clinical post. The authors recognise this major limitation of the study in the abstract and discussion.

Minor points:

1. Page 4 line 38. "Is not significantly different" from what? The point being there was no gender preference. Please re-write

Response: We have re-written the sentence to clarify the point :

"The proportion of females in a current ACF post is 48% and the proportion of males was 52%. This difference was not statistically significant (approximately 750 posts). The equal gender distribution was also apparent for those who had completed ACF posts at the time of the audit (46% female and 54% male (over 1400 posts))"

2. Page 4 line 41 - delete "extremely" significant. As this is a statistical analysis please give numbers of the HEE specialty trainee posts sampled.

Response: We have updated the sentence with the specialty trainee post numbers:

"This balanced profile is however significantly different ($p=0.0007$) to the proportion of females in specialty trainee posts (data courtesy of HEE, Nov 2015) where a significantly greater number of trainees are female (57% of 11160 current medical trainees *excluding ACFs, out of programme (OOPs), "Locum Appointments for Training (LATs) and foundation trainees, $p<0.0001$."

3. Page 4 line 58 - again the total denominator data would be useful. Please give numbers of ACF's and specialty trainees applying and numbers appointed. As this is the first report a breakdown of specialty for the ACF's would be helpful

Response: We have included the denominators to the data, revising the sentence below. Clinical non ACFs refer to medics and dentists applying who have not held an ACF.

As the overall numbers of applications to NIHR DRFs are quite small, the numbers of each specialty are also quite small and so we don't feel it would be useful to include in this section.

This has been the case since 2010. Of 190 ACF applicants, 54 were successful (28%) compared with 51 out of 273 (19%) of non-ACF clinical applicants ($p=0.01$). The overall success rate from 2010-2015 was 19% (194/1001).

4. Page 5 line 17. The survey isn't robust and doesn't really provide additional data to that reported in reference 7 which as I recall was a cross-funder containing NIHR fellows albeit at a later stage in the career (CRTF and Clinician Scientist). This might be discussed in more detail.

Response: NIHR was involved in the cross funder study and it did focus on later career stage positions. In addition to the benefits and challenges, which were certainly reflective of those collected in that earlier study, it does provide information that the issues particularly are raised at an earlier stage and are common to those attempting to navigate a clinical academic pathway. We would agree that in terms of response rates, the survey is open to non-responder bias. It does provide examples of the benefits gained by having such an award, such as the research skills and awareness that are translated into more research active clinicians, which represent a successful outcome for the award, which we have tried to emphasize in response to a related point from the other reviewer below.

5. Page 7 line 30. I didn't really understanding this paragraph at all. If the destination is purely clinical

how does this in anyway reflect the current debate as to gender balance for Academic careers?
Without data on onward career progression of this "immediate clinical" group I think this paragraph is redundant.

Response:

We accept this point and have decided to remove this paragraph.

We did start to get examples of further career progression of this group but with a lower response rate to the survey, we did not include the information in the submitted manuscript. As part of a wider review of the training programmes offered by NIHR, we are looking at further work on the gender balance of applicants and successful award holders through our schemes and therefore this fits better with the next piece of work that is ongoing.

Reviewer: 2

Jane M Blazeby

University of Bristol, UK

Please state any competing interests or state 'None declared': None declared

1. It is not clear why this paper only focuses on the impact of the ACF post and it neglects to examine ACL and AF2 posts - all of which were introduced to improve the number and quality of clinical academics. The paper also incorrectly says that an ACF post is the first step on the NIHR pathway - however the AF1/2 posts are the first step

Response: The ACF programme is a unique NIHR programme, having been created and developed by us, whereas the other posts that you mention can be funded and supported by other funders and institutions, such as the AF posts, which are HEE funded. We have therefore focussed on the ACF and the NIHR pathway rather than the academic pathway for clinicians as a whole.

We have updated the following paragraph in the introduction to try to emphasize the reason for the focus on the ACF posts:

"The NIHR Academic Clinical Fellowship (ACF) was created as a new career development scheme and is the first step on the NIHR IAT pathway and provides pre-doctoral academic training during the specialty training period for doctors and dentists."

2. The abstract could benefit from some details of what the survey was aiming to assess - and its purpose -

Response: The survey asked questions regarding current occupation and levels of research activity in order to assess career progression beyond first destination. Due to the low response rates obtained, we included just the qualitative responses to benefits and challenges faced by the award holders in this manuscript, in order to highlight some of the issues that are raised at this level. The abstract has been updated to clarify the intentions of the survey.

"Retrospective analysis of NIHR ACF career progression. This was obtained from using mixed methods including routine data collections of career destination, analysis of application rates to doctoral level fellowships and supplemented by survey information that captured the perceived benefits and challenges from previous ACFs and their current career activities".

3. The results in the abstract seem to go beyond what was likely to be surveyed - e.g. numbers of applicants and successful appointments - so the methods are unclear in this summary abstract

Response: As for comment 2 above, the abstract has been updated to clarify the methods as the survey formed just one part of the analysis. We did indeed use other analysis to look at application rates of ACFs to other NIHR schemes and collated first career destination from internal data collection and service evaluation.

4. It is unclear what a 'successful ACF is defined to be award of PhD funding'. Another key success could be that the candidate is made aware of the clinical academic world and they decide its not for them and they return to full time NHS posts - I think that more thought into some of this is needed

Response: A number of outcomes after the ACF would be considered successful, including other research training positions. This is something that we have considered, including the success of those who use an ACF as a taster in the academic world and return to their clinical positions more research-aware and who may continue to contribute to research, even if in a less formal way. We got some indication of previous ACFs doing this in the responses to the survey, we saw that some of those who selected purely clinical roles to describe their current job also indicated that they spent some of that role on research, with others selecting peer review and contributing to research of others even though they said they spent 0% of their role on research. This was reasoned by the participants by them carrying out these types of activity outside employment or not equating such work as 'research'. We would however still consider that a successful outcome. The continuation into a research training post such as a PhD is still considered the primary successful outcome as this was the main outcome when the programme was set in place. We have added the following paragraph in the discussion to emphasize this point.

"In addition to formal research training awards, a number of other outcomes would be considered successful. ACFs that do not continue onto a formal clinical academic training path and decide to return to clinical posts do so with a greater understanding of research and new skills from their ACF experience. Examples from the follow-up survey include those who indicated they are in a clinical role but also that they undertake some research activities as part of that role. Others indicated no formal time within their current role for research activities but specified that they carry out peer review and supervision of PhD students."

5. Is it possible that there is responder bias (i.e. ACFs that are successful with PhD applications have responded more than those not successful)

Response: One of the reasons we decided not to incorporate numbers from the survey results was due to the low response rate generally, with the recognition that we would likely be introducing responder bias. Whilst the profile of gender and specialty of the participants reflected those of the whole cohort, it is possible that the respondents were engaged with research and perhaps more likely to respond to a survey of its type from NIHR.

We have not amended the paper text.

6. One of the main challenges identified by ACFs of this post is the balance between clinical and academic work - I think that the discussion section of the paper could be improved if the authors considered how to tackle these issues, in particular, how they may work with NHS consultants and training pathways to make this more positive and how ACFs can be supported to negotiate these difficult issues

Response: We have added further thoughts to the discussion to expand on tackling the issue raised including new guidance that has been prepared since submitting this paper and is about to be issued for host partnerships and trainees. One of the key relationships is the Institutional Academic Training (IAT) lead role at each institution; we aim to improve the pathway by supporting their liaison with Training Programme Directors as appropriate.

“Looking to the future, NIHR will continue to monitor and track ACF post-holders along their chosen pathways and careers and evaluate the exciting impact of these posts on both clinical academic medicine and dentistry. This review and the ongoing evaluation will provide information to ensure the provision of an evidence-based trainee policy for NIHR. ACFs clearly face a range of challenges, particularly around the need to balance research, training and clinical service pressures. NIHR is therefore working with its partners to set out key principles and obligations for those in receipt of funding to ensure high quality clinical academic training is provided. NIHR have prepared a new NIHR Guide to Integrated Academic Training with clear guidance of expectations and processes for both host partnerships and trainees in order to support ACFs in the programme. ”

We would like to thank both the reviewers and editor for their time and helpful comments to improve the paper.

VERSION 2 – REVIEW

REVIEWER	Paul Stewart University of Leeds
REVIEW RETURNED	25-Mar-2017

GENERAL COMMENTS	I am happy that my queries have been addressed - accepting the limitations of the study as outlined.
--

REVIEWER	Jane M Blazeby University of Bristol, UK
REVIEW RETURNED	05-Apr-2017

GENERAL COMMENTS	I have no further comments. The response to my comments (and those of the other reviewer) are detailed and answer the issues we raised.
---